

# Graphics Algorithm for Deriving Atmospheric Boundary Layer Heights from CALIPSO Data

**Boming Liu[1], Yingying Ma[1,2*], Jiqiao Liu[3*], Wei Gong[1,2], Wei Wang[4] and Ming Zhang[1]**

[1]  State Key Laboratory of Information Engineering in Surveying, Mapping and Remote Sensing (LIESMARS),
    Wuhan University, Wuhan 430079, China

[2]  Collaborative Innovation Centre for Geospatial Technology, Wuhan 430079, China

[3]  Shanghai Institute of Optics and Fine Mechanics, Chinese Academy of Sciences, Shanghai, 201800

[4]  School of Geoscience and Info-Physics, Central South University, Changsha, 410083

**∗**  Correspondence: yym863@whu.edu.cn

**Abstract:** The atmospheric boundary layer is an important atmospheric feature that affects environmental health and weather forecasting. In this study, we proposed a graphics algorithm for the derivation of atmospheric boundary layer height (BLH) from the Cloud-Aerosol Lidar and Infrared Pathfinder Satellite Observations (CALIPSO) data. Owing to the differences in scattering intensity between molecular and aerosol particles, the total attenuated backscatter coefficient 532 and attenuated backscatter coefficient 1064 were used simultaneously for BLH detection. The proposed algorithm transformed the gradient solution into graphics distribution solution to overcome the effects of large noise and improve the horizontal resolution. This method was then tested with real signals under different horizontal smoothing numbers (1, 3, 15 and 30). The algorithm provided a reliable result when the horizontal smoothing number was greater than 5. Finally, the results of BLH obtained by CALIPSO data were compared with the results retrieved by the ground-based Lidar and radiosonde (RS) measurements. Under the horizontal smoothing number of 15, 9 and 3, the correlation coefficients between the BLH derived by the proposed algorithm and ground-based Lidar were 0.72, 0.72 and 0.14, respectively, and those between the BLH derived by the proposed algorithm and radiosonde measurements were 0.59, 0.59 and 0.07. When the horizontal smoothing number was 15 and 9, the CALIPSO BLH derived by the proposed method demonstrated a good correlation with ground-based Lidar and RS. This finding indicated that the proposed algorithm can be applied to the CALIPSO satellite data with 3 and 5 km horizontal resolution.

**Keywords:** Aerosol; Lidar; Radiosondes; Boundary layer height; CALIPSO

## 1. Introduction

The atmospheric boundary layer is the layer of the Earth's surface atmosphere which is closely related to human activities (Bonin et al. 2013; Reuder et al. 2009; Flamant et al. 1997). It plays a crucial role in regional environmental health and is important in weather forecasting model (Liu et al. 2018a; Leventidou et al. 2013). Meanwhile, the heating process of solar radiation for the surface are also achieved through boundary layer dynamics (Yang et al, 2013). Furthermore, atmospheric activity in the boundary layer affects the propagation of cloud nuclei and pollutant dispersion (Lange et al, 2014). Therefore, the boundary layer height (BLH) is essential to environmental health and human activity and thus must be accurately monitored.



Various detection technologies are currently used for BLH observation, including optical (Lidar, ceilometers) and electromagnetic (radiosondes, Doppler radar) remote sensing (Seibert et al. 2000; Sawyer et al. 2013). The RS was the most common measurement instrument used for detecting the vertical profiles of meteorological parameters (Hennemuth et al. 2006). The BLH can be derived from the thermodynamic profiles measured by the

RS (Holzworth et al. 1964). However, the observation time of RS is discontinuous. That is, RS is typically launched routinely twice a day or from four to eight times daily during field experiments (Holzworth et al. 1967). Moreover, the spatial coverage of RS sites is usually sparse to capture BLH spatial variability. The ground-based Lidar system is an active remote sensing equipment, which can provide aerosol extinction profile with a high spatial resolution (Huang et al, 2010). This system has been widely employed for the study of the optical and

physical properties of atmospheric aerosols (Melfi et al. 1985). Lidar systems can detect the BLH from the aerosol vertical profile (Li et al. 2017). However, owing to expensive price and maintenance costs, the spatial coverage of ground-based Lidar remains poor.

The CALIPSO is the only space-borne Lidar in operation in the world (Winker et al. 2007, 2009; Liu et al. 2015). CALIPSO provides the vertical distributions of clouds and aerosols with high vertical resolution and offers

a significant potential for the estimation of global BLHs from space (Mamouri et al. 2009). The major methods of deriving BLH from CALIPSO data include the wavelet covariance transform (WCT) and maximum standard deviation (MSD) methods (McGrath-Spangler et al. 2012; Brooks et al. 2003). Through these methods, the BLH can be determined by using the vertical profile of aerosol. The MSD method determines the BLH from CALIPSO as the lowest occurrence of a local maximum in the standard deviation of backscatter profile collocated with a

maximum in the backscatter itself (Jordan et al. 2010). The WCT method searches the local maximum with a coherent scale and defines the height of maximum value as BLH (Davis et al. 2000). These methods have been widely used for BLH derivation. However, due to the low signal-to-noise ratio (SNR) of CALIPSO data, these methods can be applied to the real signals only when the horizontal smoothing number is large. The CALIPSO provided the total attenuated backscatter coefficient with a horizontal resolution of 1/3 km (Winker et al. 2009). In

particular, the signals reaching the CALIPSO Lidar from low altitudes can possess significant noise due to the long travel distance of attenuated backscatter. The large noise conceals the gradient value at the top of boundary layer when the horizontal smoothing number was small. Thus, obtaining the BLH by using the WCT and MSD methods is difficult. Therefore, a horizontal smoothing method is necessary for improving the SNR of satellite data (Guo et al. 2016). Zhang et al. (2016) obtained a 5 km horizontal smoothing profile by averaging 15 CALIOP vertical

profiles to retrieve BLH results. Su et al. (2017) retrieved BLHs from a 7 km horizontal smoothing (horizontal smoothing number = 21) CALIPSO data to minimise the influence of outliers. In this manner, the noise of satellite data can be effectively restrained, and the BLH results can be obtained from CALIPSO. However, this method sacrifices the horizontal resolution of CALIPSO detection.

In this research, we proposed a graphics distribution method (GDM) for deriving the BLH from CALIPSO

data and preventing significant reduction of horizontal resolution. The total attenuated backscatter coefficient 532 ($TAB_{532}$) and attenuated backscatter coefficient 1064 ($AB_{1064}$) were used for the construction of two-dimensional graphics distribution, which was used for BLH derivation. The GDM algorithm was then tested with real signals under different horizontal smoothing numbers. Finally, the results of BLH obtained by CALIPSO data were compared with those retrieved by the ground-based Lidar and RS measurements during January 2013 to December

40 2017.



## 2. Materials

### 2.1 Study Aera

The CALIPSO, ground-based Lidar, and RS were used for calculation of BLHs over Wuhan, a megacity close to the Han and the Yangtze River. Wuhan is one of the most densely populated and industrialised region over central China (Liu et al. 2018b; Zhang et al. 2017).

In the Wuhan area, the ground-based Lidar stations are located at the State Key Laboratory of Information Engineering in Surveying, Mapping and Remote Sensing, Wuhan University (30 ′32'N, 114 ′21'E) (Liu et al. 2017). The RS measurements station (30°37'N, 114°08'E) operated by the Wuhan meteorological bureau routinely launches RS at 8:00 and 20:00 local time (LT) daily (Liu et al. 2018a). Fig. 1 shows the geographic distributions of the ground-based Lidar and RS station. The black point and blue triangle represent the ground-based Lidar and RS station, respectively. The black line represents the track of CALIPSO satellite. The trajectory of CALIPSO does not completely coincide, but the distance between CALIPSO and ground-based Lidar stations is within 50 km.

### 2.2 Ground-based Lidar Data

A ground-based Lidar system was used for the detection of the atmospheric vertical profiles in Wuhan (Wei et al. 2015). The Lidar system uses a pulsed Nd: YAG laser with 532 nm wavelength. The pulse rate of the laser was 20 Hz, and the laser energy was 150 mJ. The vertical resolution of the system was 3.75 m, and the acquisition frequency of the system was 20 Hz. Additional details of this Lidar system can be found in previous studies. Given that the Lidar signal is susceptible to the noise of background light during daytime, the Lidar system was employed at night from 19:00 to 07:00 local time. The ideal profile fitting method proposed by Steyn et al. (1999) is an effective method for delineating stable boundary layers. As this method can be successfully used for obtaining the BLH in ground-based Lidar research, an ideal profile fitting method was used.

### 2.3 CALIPSO Data

The CALIOP satellite is the first space-borne Lidar optimised for aerosol and cloud profiling, which has 532 nm channel (parallel and perpendicular polarisation) and 1064 nm channel (Liu et al. 2009). This satellite can provide the total attenuated backscatter coefficient 532 and attenuated backscatter coefficient 1064 with a horizontal resolution of 1/3 km and vertical resolution of 30 m. Attenuated backscatter data (Level 1B) were used for testing the proposed algorithm. The cycle time of CALIPSO across the central China region is 16 days, and the crossing time of the satellite in Wuhan is 13:10 and 02:10 local time. The nighttime data were employed for this analysis for the matching of the ground-based data, and cases with cloud and dust were removed in this study.

### 2.4 Radiosonde Measurements Data

The RS data were provided by the Bureau of Meteorology at Wuhan site, which is 23 m above sea level and 30 km northwest from the Lidar site. The RS was launched twice a day at 8:00 (LT) and 20:00 (LT). The RS data from 20:00 (LT) were selected to calculate the BLH and match the satellite data (Pal et al. 2013). The vertical profiles of the mean horizontal wind speed and potential temperature were used to determine the BLH following the method described in Liu and Liang because the construction of nighttime boundary layer is complicated (Liu et al. 2010). Moreover, due to the mismatched time of RS data, the BLH estimated from RS measurements data





cannot be regarded as 'truth'; thus, the estimated BLH is jointly used with the ground-based Lidar for validating CALIPSO results.

**3. Methodology**

*3.1. Method*

Previous studies reported that the different particles are distributed in different vertical heights (Liu et al. 2018c; Sugimto et al. 2002). Most of the particles above the boundary layer are molecular particles, and the particles below the boundary layer are mainly aerosol particles, as shown in Figs. 2a and 2c, respectively. Therefore, we proposed a dual-wavelength algorithm that determines BLH on the basis of two-dimensional graphical distribution. The total attenuated backscatter coefficient 532 ($TAB_{532}$) and attenuated backscatter

coefficient 1064 ($AB_{1064}$) were used to construct the two-dimensional graphical distribution. The specific steps are as follows:

Firstly, the $TAB_{532}$ and $AB_{1064}$ were employed for the construction of the sample sequence $X(z)$. As shown in Figs. 2b and 2d, the $TAB_{532}$ and $AB_{1064}$ represent the aerosol vertical profile at 532 and 1064 wavelength measured by CALIPSO, respectively. The $X(z)$ can be expressed as:

$$\left[ X(z) \right] = \left[ TAB_{532}(z), AB_{1064}(z) \right] \tag{1}$$

where $z$ stands for the altitude of sample points; $X(z)$ represents the coordinates of the sample point at the altitude of z; $TAB_{532}(z)$ and $AB_{1064}(z)$ represent the total attenuated backscatter (532 nm) and attenuated backscatter (1064 nm) value of the sample point at the altitude of z, respectively.

The sample sequence $X(z)$ is shown in Fig. 3a. The colour bar is the altitude of sample points. The figure

shows that $TAB_{532}$ and $AB_{1064}$ of blue points (the particles below the boundary layer) were larger than those of the red points (the particles above the boundary layer). According to this two-dimensional distribution, the sample sequence $X(z)$ can be divided into two categories.

The k-means method was used for the classification of the sample sequence. Two centroid points (u1, u2) were randomly selected from the sample sequence. For each sample point of sample sequence $X(z)$, the cluster C

belonging to is calculated as follows:

$$c(z) = \arg\min_{j} \left\| X(z) - u_j \right\|^2 \tag{2}$$

where $C(z)$ represents the cluster of sample point at the altitude of z, and $u_j$ is the centroid of cluster j ($u_1$ or $u_2$). For each cluster j, the centroid $u_j$ is recalculated as follows:

$$u_j = \frac{\sum_{i=1}^{m} 1\{c(i) = j\} x(i)}{\sum_{i=1}^{m} 1\{c(i) = j\}} \tag{3}$$

Eqs. (3) and (4) are repeated until the centroids ($u_1$ and $u_2$) converge. The sample sequence is divided into two categories after the convergence. As shown in Fig. 3b, $cluster_2$ (blue points) indicates the aerosol particles below the boundary layer, and $cluster_1$ (red points) is the molecular particles above the boundary layer. Black cross represents centroid points. Meanwhile, the categories sequence $f(z)$, which changes with height, can be obtained and expressed as:



$$f(z) = \begin{cases} 1, & z \in cluster_1 \\ 2, & z \in cluster_2 \end{cases} \tag{4}$$

where $f(z)$ is the category of sample point at the altitude of $z$. The noise points would affect the classification results due to the large noise of satellite data. Therefore, the noise points on the categories sequence must be eliminated. The noise point was determined by comparing two points near the point. If the two points above and below this

point belong to the same class, then this point should also belong to this category. The noise point can be filtered by:

$$f(z) = f(z-m), \quad if: f(z-m) = f(z+m) \tag{5}$$

where $m$ represents the multiple of the vertical resolution, and the different values can be selected at different noise levels. When the horizontal smoothing number is small and the signal noise is large, the value of $m$ can be set as 2;

and when the horizontal smoothing number is large, the value of $m$ can be set as 1. Hence, the noise points were removed, and the new categories sequence $F(z)$ was obtained as follows:

$$F(z) = \begin{cases} 1, & z > BLH \\ 2, & z < BLH \end{cases} \tag{6}$$

where $F(z)$ represents the category of the sample point at the altitude of $z$. $BLH$ indicates the BLH result. Fig. 3c shows the category sequence $F(z)$, which contains the height information and shows evident variation at the top of

boundary layer. Therefore, the maximum gradient of the categories sequence $F(z)$ is the top point of boundary layer. The $BLH$ can be calculated by searching the maximum gradient, which can be expressed as:

$$BLH = \left| d\left[F(z)\right] \right|_{max} \tag{7}$$

Following this process, the BLH was obtained based on the two-dimensional distribution of particles.

### 3.2. Error analysis

Fig. 4 shows the flowchart of the GDM algorithm. Four calculation steps are available: establishing the sample sequence, particle clustering, filtering noise points, and maximum gradient searching. The error of input parameters is the main factor affecting the accuracy of the algorithm because these steps are quantitative calculations. According to the official description, the uncertainty of backscatter coefficient was 20%–30% (Winker et al. 2009). The total attenuated backscatter at 532 nm wavelength and the attenuated backscatter at 1064

nm wavelength were measured from CALIPSO. Therefore, the error of input parameters was 20%–30%. The error of the BLHs derived by the GDM algorithm is approximately 20%–30%. In addition, it need to note that this method cannot be applied to low cloud and dust cases, because the boundary of cloud or dust would be misclassified to BLH.

### 4. Results

The GDM algorithm was applied to the CALIPSO data acquired from January 2013 to December in 2017. After removing the cases with cloud and dust, the number of residual CALIPSO data over Wuhan area was 49. In addition, the results of BLH were compared with those retrieved by the ground-based Lidar and RS measurements.

### 4.1. Testing with real signals



Fig. 5 shows the case study of CALIPSO data with different horizontal smoothing numbers on 4 October 2013 over Wuhan area. Figs. 5a, 5b, 5c and 5d represent the vertical profile of $TAB_{532}$ derived from CALIPSO profile with a horizontal smoothing number of 1, 3, 15 and 30, respectively, and their BLH result was 1020, 980, 980 and 980 m, respectively. Fig. 5a shows the vertical profile of $TAB_{532}$ with a horizontal smoothing number of 1, in which the noise of satellite data was large. Such noise produced discrete sample sequence distribution. However, the category sequence and the BLH result (1010 m) can still be obtained. As shown in Figs. 5b, 5c and 5d, the noise of satellite data was reduced with the increase in horizontal smoothing number. Moreover, the vertical profile of $TAB_{532}$ derived from CALIPSO profile was gradually becoming smooth. Such transformation resulted in significantly compact distribution of sample sequences (Fig. 5d), which were conducive to the classification of sample points. The categories sequence was easily obtained from the classification calculation, and the result of the BLH converged to 980 m. In this case, the GDM algorithm can obtain the BLH result under different horizontal smoothing numbers.

Fig. 6 shows the case study of CALIPSO data with different horizontal smoothing numbers on 12 February 2015 over Wuhan area. The BLH result of Figs. 6a, 6b, 6c and 6d were 532, 1280, 1370 and 1370 m, respectively. As shown in Fig. 6a, when the horizontal smoothing number was 1, the high noise of CALIPSO mixed together the sample points at different heights. In this condition, the categories sequence cannot accurately distinguish between molecular and aerosol particles. Therefore, an inaccurate BLH result was obtained under this condition. When the horizontal smoothing number was added to 3 (Fig. 6b), the distribution of sample sequences significantly improved, and the obtained BLH result was 1280 m. Figs. 5c and 5d show the vertical profile of $TAB_{532}$ with the horizontal smoothing number of 15 and 30, respectively, in which the distribution of sample sequences gradually became compact. The result of the BLH converged to 1370 m. This result indicates that the GDM algorithm cannot be applied to the data with horizontal smoothing number of 1 in this case, but it can provide a relatively reliable result when the horizontal smoothing number was greater than 3.

The relationship between the horizontal smoothing number and BLH was investigated to determine the convergence of the BLH results. Fig. 7 shows the relationship between the horizontal smoothing number and BLH under different cases. Fig. 7a shows the case study on 4 October 2013. The result of the BLH converges to 980 m when the horizontal smoothing number was greater than 2. Fig. 7b shows the case study on 12 February 2015. The result of the BLH converged to 980 m when the horizontal smoothing number was greater than 4. These results indicate that the PDM algorithm was not applied to the satellite data when the horizontal smoothing number was extremely small. However, this algorithm can provide a reliable result when the horizontal smoothing number is greater than 5.

### 4.2. Comparison with other algorithms

In this section, we compare the results of BLH obtained by CALIPSO data with those retrieved by the ground-based Lidar and RS measurements to verify the stability of the algorithm. The number of the ground-based Lidar and RS data matching CALIPSO data were 21 and 49, respectively. The results of BLH calculated by the MSD method were used as a reference.

Figs. 8a and 8c show the total attenuated backscatter at 532 nm plot from CALIPSO on 7 October 2014 under the horizontal smoothing number 15 and 9, respectively. The black and blue line represent the BLH results calculated by GDM algorithm and MSD method, respectively. The red circle stands for the BLH result from ground-based Lidar. Fig. 8b shows the corresponding vertical profile of $TAB_{532}$ derived from CALIPSO profile



over Wuhan area under the horizontal smoothing number 15. The BLH results calculated by GDM algorithm, MSD method and ground-based Lidar were 1220, 980 and 1250 m, respectively. Fig. 8b shows the corresponding vertical profile of $TAB_{532}$ under the horizontal smoothing number 9. The BLH results calculated by GDM algorithm, MSD method and ground-based Lidar were 1220, 770 and 1250 m, respectively.

Figs. 9a, 9b and 9c show the correlation coefficients between the BLH derived from CALIPSO and ground-based Lidar under the horizontal smoothing numbers of 15, 9 and 3. The red and blue points represent the BLH calculated by GDM algorithm and MSD method, respectively. The correlation coefficients between the BLH derived by GDM algorithm and ground-based Lidar were 0.72, 0.72 and 0.14 under the horizontal smoothing number of 15, 9 and 3, respectively. Meanwhile, the correlation coefficients between the BLH derived by MSD

method and ground-based Lidar were 0.7, 0.54 and 0.27. The correlation between BLH derived from RS measurements and CALIPSO is shown in Figs. 9d, 9e and 9f. Under the horizontal smoothing number of 15, 9 and 3, the correlation coefficients between the BLH derived by GDM algorithm and RS measurements were 0.59, 0.59 and 0.07, respectively, and the correlation coefficients between the BLH derived by MSD method and RS measurements were 0.54, 0.42 and 0.03, respectively. These results indicate that the performance of GDM

algorithm was similar to the MSD method when the horizontal smoothing number was large (15). When the horizontal smoothing number was 9, the performance of GDM algorithm was superior to the MSD method. Moreover, the GDM algorithm and MSD method show a poor performance when the horizontal smoothing number was small (3).

### 5. Discussion

The CALIPSO satellite is a powerful tool for monitoring the vertical distribution of clouds and aerosols, which offers a significant potential for the estimation of global BLHs from space (Winker et al. 2007, 2009). Moreover, a horizontal smoothing method was used to improve the SNR of satellite data due to its large noise (Guo et al. 2016; Zhang et al. 2016; Su et al. 2017). However, this method considerably sacrificed the horizontal resolution of CALIPSO detection. A graphics algorithm was proposed to determine the BLHs from CALIPSO data

and overcome this problem.

     The total attenuated backscatter coefficient 532 and attenuated backscatter coefficient 1064 were used to construct the two-dimensional graphics distribution, as shown in Fig. 3a. The extremum and negative points can be filtered through this graphics distribution. The sample sequence was then classified by the k-means method, and the categories sequence was obtained (Fig. 3b). When the horizontal smoothing number was different, the degree

of noise was also different. When the noise was large, the noise point which was above the boundary layer and may be classified below the boundary layer, thereby significantly affecting the accuracy of categories sequence. Therefore, the noise points were removed again, and the new categories sequence was obtained (Fig. 3c). The BLH result can be determined from the new categories sequence by maximum gradient search (Fig. 3d). The advantage of the GDM algorithm is that this algorithm transforms the gradient solution into graphics distribution solution.

The multiple gradient values in the backscatter coefficient profile can be understood as the extremely dispersed distribution of the particles. According to the graphic classification, the influence of noise gradient can be avoided, and a reliable BLH result can be obtained.

     The test results of GDM algorithm are shown in Figs. 5, 6 and 7. These results indicate that the GDM algorithm can be applied to the satellite data when the horizontal smoothing number is small. However, when the

horizontal smoothing number is below 5, the large noise affects the distribution of the sample sequence, and



obtaining the BLH by graphic classification is difficult. Regarding the performance of algorithm, as shown in Figs. 8 and 9, the performance of the GDM algorithm was similar to that of the MSD algorithm when the horizontal smoothing number was large. This finding can be attributed to the noise of satellite data, which produced the evident gradient of aerosol concentration when effectively restrained by the horizontal smoothing method. Thus,

both the algorithms can accurately detect the BLH. However, with the decrease in the number of horizontal smoothing, a difference was observed between the GDM and MSD algorithms with respect to performance. When horizontal smoothing number was small (9), the noise of satellite data was ineffectively controlled, thereby resulting in multiple gradients in the vertical direction. The MSD algorithm failed to obtain the effective BLH from the multiple gradient values. However, the GDM algorithm can still detect the BLH based on the graphics

distribution, overcome the effect of multiple gradient values and accurately identify the BLH. Therefore, the GDM algorithm can deal well with the CALIPSO data with a small horizontal smoothing number.

### 6. Conclusions

    We proposed a graphics algorithm to obtain the BLHs from CALIPSO data. The following four calculation steps were used: establishing the sample sequence, particle clustering, filtering noise points and maximum gradient

searching. The $TAB_{532}$ and $AB_{1064}$ were used for the construction of the two-dimensional graphics distribution. Based on the graphics distribution of atmospheric particulate, the k-means method was used for the classification of the sample sequence and acquisition of the BLH. The algorithm was then applied to the real signals with different horizontal smoothing numbers for the evaluation of the algorithm's performance. The results indicate that the performance of GDM algorithm was poor when the horizontal smoothing number was extremely small (such

as 1 to 3), although it can provide a reliable result when the horizontal smoothing number was greater than 5. Finally, the results of BLH obtained by CALIPSO data were compared with those retrieved by the ground-based Lidar and RS measurements from January 2013 to December 2017. Notably, when the horizontal smoothing number was extremely large (above 15), the performance of the GDM algorithm was similar to that of the MSD method. It indicated that the 5 km horizontal resolution CALIPSO data (the horizontal smoothing number of 15)

was suitable for both GDM and MSD method to derive the BLH. Moreover, the correlation coefficients between the BLH derived by the GDM method and ground-based Lidar were superior to those between the BLH derived by the MSD method and ground-based Lidar when the horizontal smoothing number was 9. This finding indicates that the performance of the GDM algorithm is superior to that of the MSD method when the 3 km horizontal resolution CALIPSO data was used. Overall, the CALIPSO BLH derived by GDM method is reasonably

consistent with ground-based Lidar and RS. The MSD algorithm can derive the BLH effectively from the 3 km and 5 km horizontal resolution CALIPSO data.

**Acknowledgments:** This work was supported by the National Key R&D Program of China (2017YFC0212600), and the Haze Program of the Wuhan Technological Bureau (2017CFB404), and the National Natural Science Foundation of China (Program No. 41127901 and No. 41627804).

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

25

30



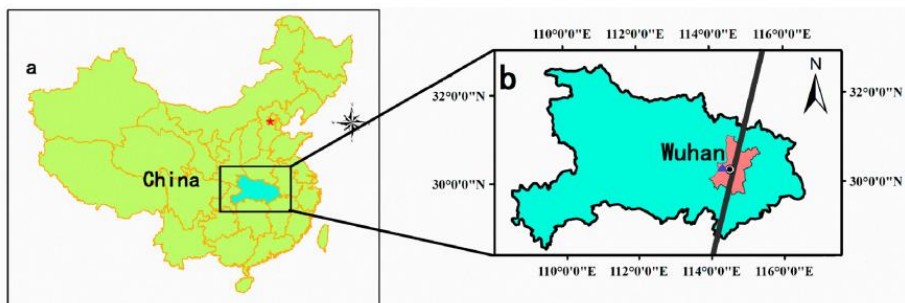

**Figure 1. Geographic distributions of the ground-based Lidar and RS measurements station. The black point and blue triangle represent the ground-based Lidar and radiosonde measurements station, respectively. The black line represents the track of CALIPSO satellite.**





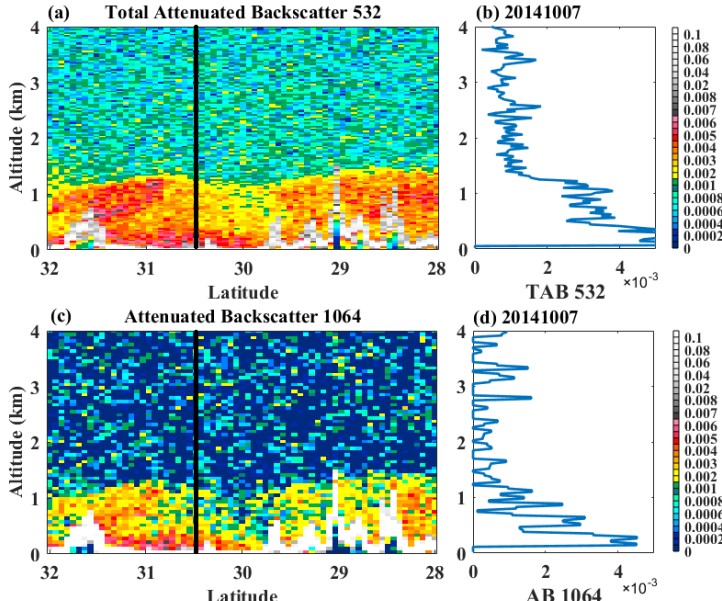

**Figure 2. (a) Total attenuated backscatter at 532 nm wavelength (TAB$_{532}$) and (c) the attenuated backscatter at 1064 nm wavelength (AB$_{1064}$) plot from CALIPSO on 7 October 2014. (b) and (c) indicate the corresponding vertical profile of TAB$_{532}$ and AB$_{1064}$ derived from CALIPSO profile over Wuhan area, respectively. The number of horizontal smoothing is 20.**




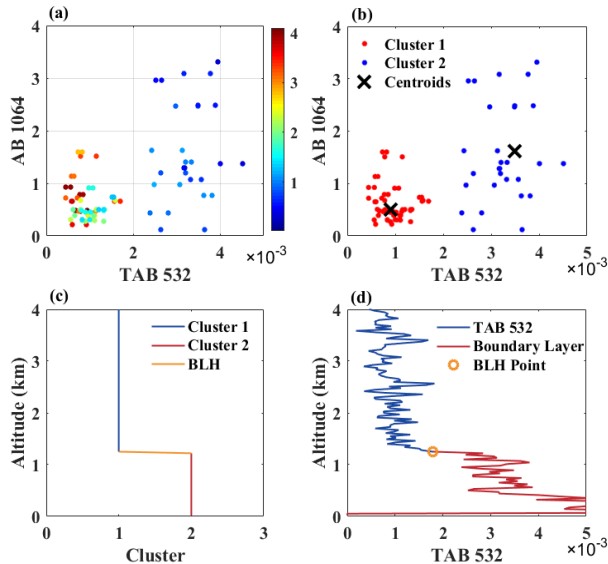

**Figure 3.  Case study over Wuhan area on 7 October 2014. (a) Scatter plots of TAB$_{532}$ and AB$_{1064}$. The color bar shows the altitude of sample point. (b) Classification results. The red and blue point represent the cluster 1 and 2, respectively. The black fork represents the centroid of the cluster. (c) The sequence of category F(z). (d) The result of the case analysis. The orange circle represents the result of BLH.**





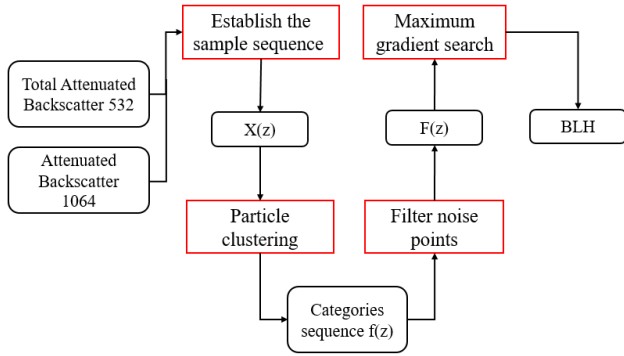

**Figure 4. Flowchart of the GDM algorithm. The red box indicated the four calculation steps.**

25





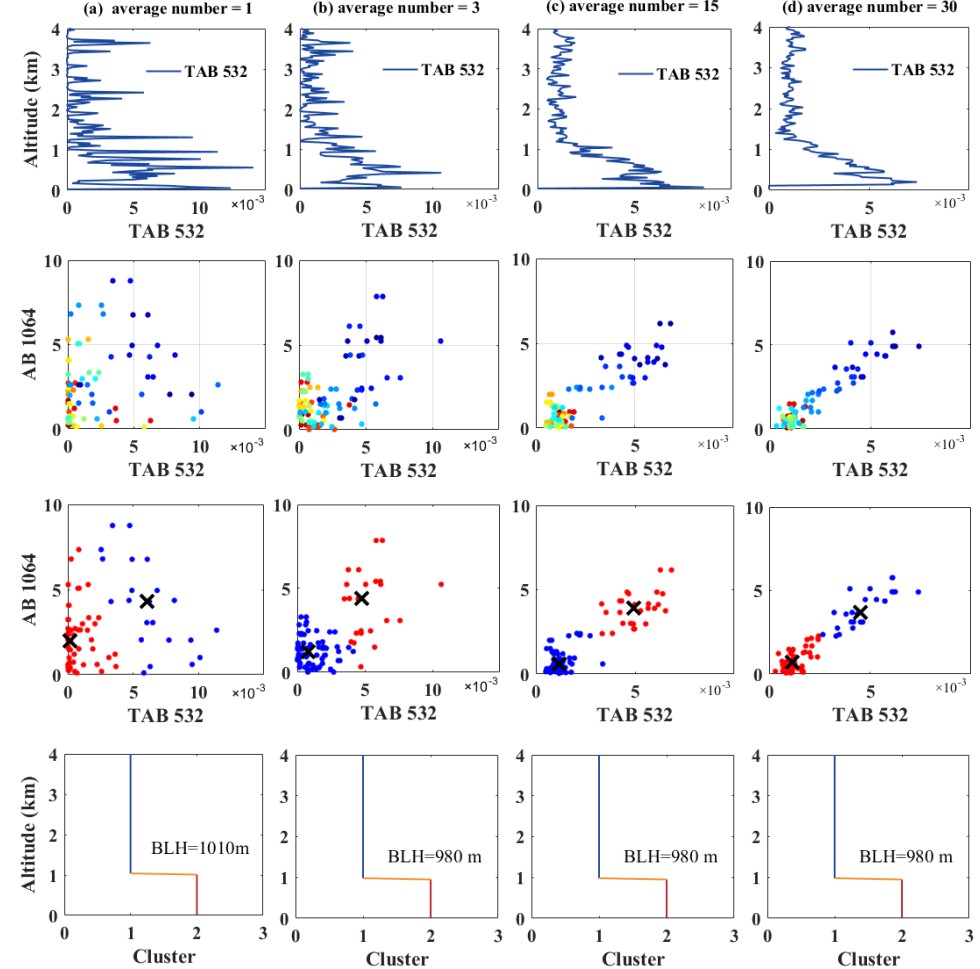

**Figure 5. Case study of CALIPSO data with different horizontal smoothing numbers on 4 October 2013 over Wuhan area. (a) average number = 1, (b) average number = 3, (c) average number = 15 and (d) average number = 30. The blue line represents the vertical profile of TAB$_{532}$ derived from CALIPSO data, the Black cross represents the centroid of the cluster and the orange horizontal line represents the BLH result.**





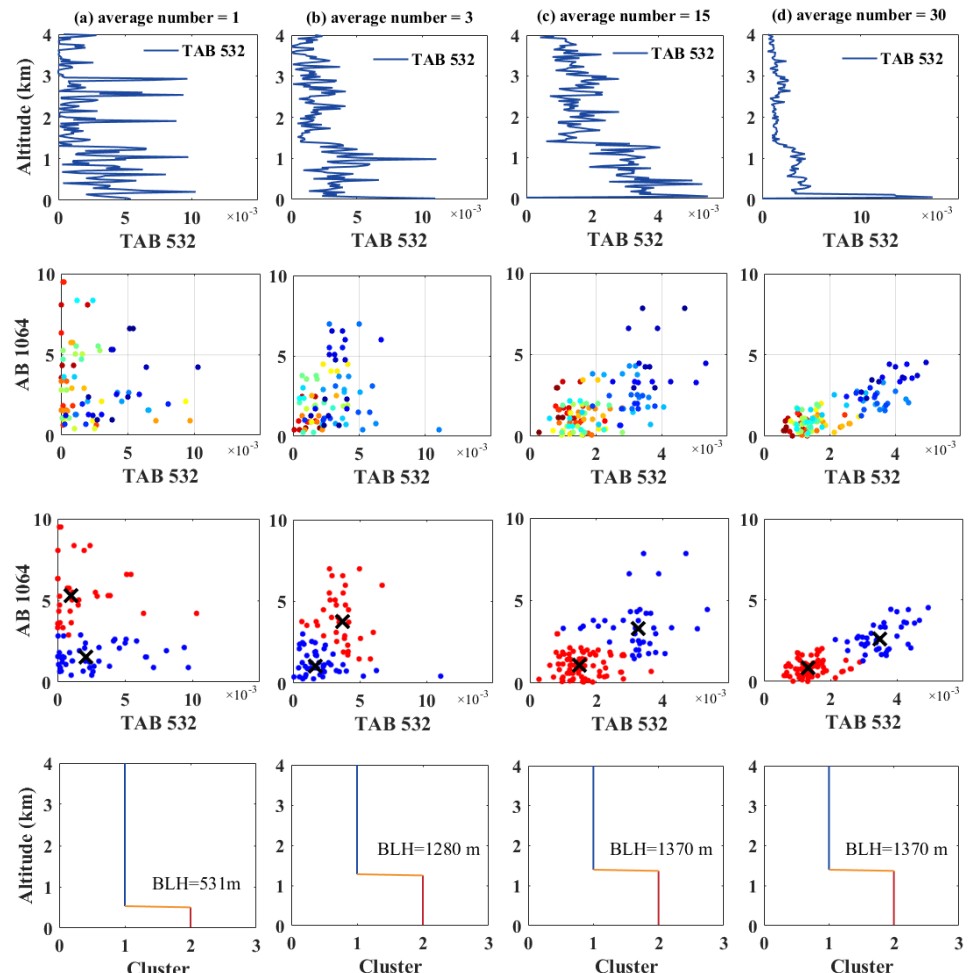

**Figure 6 Case study of CALIPSO data with different horizontal smoothing numbers on 12 February 2015 over**

**Wuhan area. (a) average number = 1, (b) average number = 3, (c) average number = 15 and (d) average number =**

**30. The blue line represents the vertical profile of TAB$_{532}$ derived from CALIPSO data, the black cross represents**

**the centroid of the cluster and the orange horizontal line represents the BLH result.**



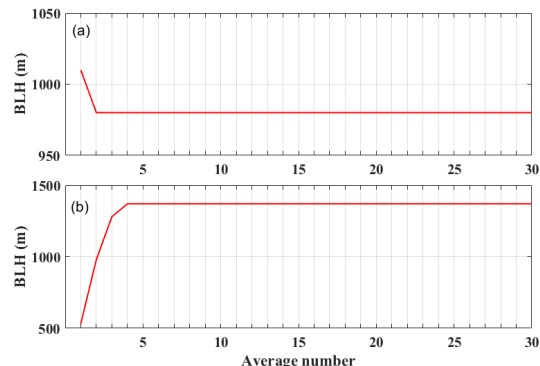

**Figure 7. Relationship between the horizontal smoothing number and BLH under different cases: (a) 4 October 2013 and (b) 12 February 2015.**

25





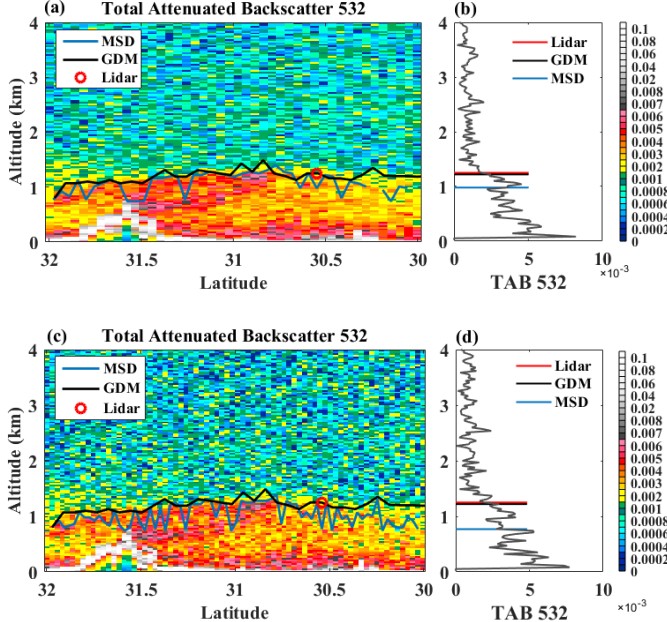

**Figure 8. Total attenuated backscatter at 532 nm wavelength (TAB$_{532}$) plot from CALIPSO on 7 October 2014**

**under the horizontal smoothing number (a) 15 and (c) 9. The corresponding vertical profile of TAB$_{532}$ derived**

5      **from CALIPSO profile over Wuhan area under the horizontal smoothing number (b) 15 and (d) 9.**




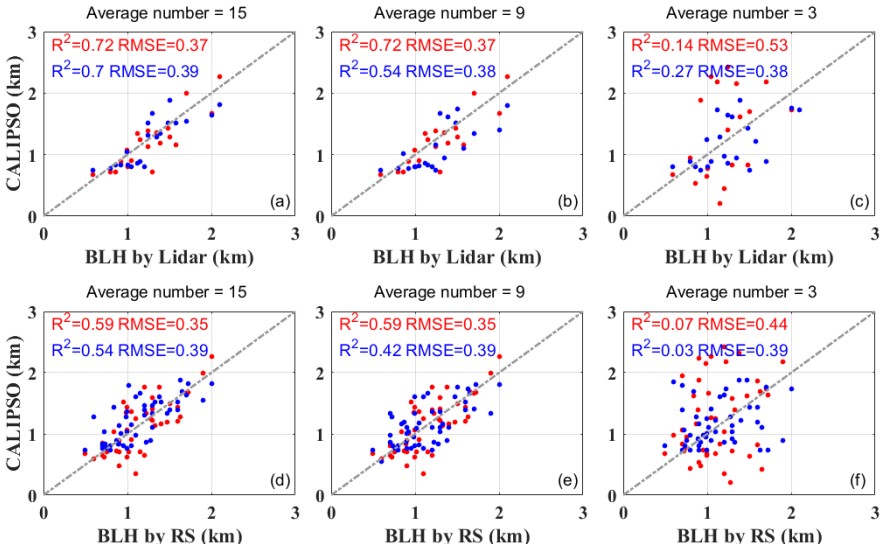

**Figure 9 Correlation of BLH derived from CALIPSO and ground-based Lidar under the horizontal smoothing number of (a) 15, (b) 9 and (c) 3. The correlation of BLH derived from CALIPSO and RS measurements under the horizontal smoothing number of (d) 15, (e) 9 and (f) 3. The red and blue points represent the BLH calculated**

5        **by GDM algorithm and MSD method, respectively.**