# Peer review of "Graphics Algorithm for Deriving Atmospheric Boundary Layer Heights from CALIPSO Data"

_Atmospheric Measurement Techniques, 2018_

## Short Comment (SC1) · 13 Jul 2018

This manuscript proposed a graphics algorithm for determining BLH from CALIPSO. However, some part of this paper should be carefully considered:

1. The author claimed that they use nighttime data of CALIPSO (0210LT), and calculate the BLH from RS following Liu and Liang (2010). Based on Liu and Liang's paper, the majority of BLHs at 0200 LT is less than 500m (such results from 14 major field campaigns). Nonetheless, in this study, the BLHs at 0210LT from CALIPSO and Lidar are all above 500m (mostly higher than 1km).

2. This study shows that the R2 between the BLHs from CALIPSO at 0210LT and RS at 2000LT is 0.59 (i.e. Figure 9). As we known, the BLH has strong diurnal variances,

and the BLHs at 2000LT (previous day) and 0200LT should have considerable differences. Although I am not sure about CALIPSO's performance, the R2 of 0.59 (R=0.77) between the BLHs at 2000LT and 0210LT may be questionable.

3. The authors claimed that the cycle time of CALIPSO is 16 days with removals of cloud cases. Given that the night cloud fraction is ~60% over central China (King et al., 2013), the available CALIPSO sampling for matching Lidar is limited. The author should describe the continuous observation period for Lidar, which may be longer than 2-year.

References

Liu, S. and Liang, X.Z., 2010. Observed diurnal cycle climatology of planetary boundary layer height. Journal of Climate, 23(21), pp.5790-5809.

King, M.D., Platnick, S., Menzel, W.P., Ackerman, S.A. and Hubanks, P.A., 2013. Spatial and temporal distribution of clouds observed by MODIS onboard the Terra and Aqua satellites. IEEE Transactions on Geoscience and Remote Sensing, 51(7), pp.3826-3852.
* * *

---

## Author Comment (AC1) · 24 Jul 2018

Dear D. Philips:

Thank you very much for your guidance and advice. We carefully read your suggestions, and revised the manuscript in accordance with your comments.

1. The reviewer's comment: The author claimed that they use nighttime data of CALIPSO (0210LT), and calculate the BLH from RS following Liu and Liang (2010). Based on Liu and Liang's paper, the majority of BLHs at 0200 LT is less than 500m (such results from 14 major field campaigns). Nonetheless, in this study, the BLHs at 0210LT from CALIPSO and Lidar are all above 500m (mostly higher than 1km).

The authors' Answer: Thank you very much for your suggestion and guidance. As your

said, the nocturnal boundary layer height retrieved from RS should be less than 500m. It is due to the RS define the top of the inversion layer as the boundary layer height. In the nighttime, the structure of boundary layer is divided into a stable layer and a residual layer. The top of the inversion layer was close to the top of stable layer. But the Lidar system obtained the boundary layer height based on the aerosol scattering profile. If the aerosol loading in the residual layer is large, the top of residual layer would be identified as the boundary layer height by Lidar. After our experiment, we found that the CALIPSO system was hard to identify the top of stable layer in nighttime. Therefore, the top of residual layer was defined as the boundary layer height in CALIPSO and Lidar system. It leads to that the BLHs from CALIPSO and Lidar are all above 500m. About this question, more details would be added in the 3.2 section (Error analysis) to avoid misleading readers. Meanwhile, overcoming the effect of the residual layer on CALIPSO is our future work.

2. The reviewer's comment: This study shows that the R2 between the BLHs from CALIPSO at 0210LT and RS at 2000LT is 0.59 (i.e. Figure 9). As we known, the BLH has strong diurnal variances, and the BLHs at 2000LT (previous day) and 0200LT should have considerable differences. Although I am not sure about CALIPSO's performance, the R2 of 0.59 (R=0.77) between the BLHs at 2000LT and 0210LT may be questionable.

The authors' Answer: Thank you very much for your suggestion and guidance. About this point, we agree with your viewpoint that the correlation coefficient between the BLHs at 2000LT and 0210LT may be questionable. Due to the BLHs from CALIPSO at 0210LT was the top of residual layer, but the BLHs from RS at 2000LT was the top of inversion layer, which indicated the top of mixing layer in daytime. The high correlation coefficient between them could be a coincidence. Moreover, we have mentioned in the study that "Due to the mismatched time of RS data, the BLH estimated from RS measurements data cannot be regarded as 'truth'; thus, the estimated BLH is jointly used with the ground-based Lidar for validating CALIPSO results." Therefore, this comparison is just a reference. If necessary, we would choose to compare with the RS data at 0800LT (indicated the top of stable layer in nighttime), or not compare with the RS data to avoid misleading readers.

3. The reviewer's comment: The authors claimed that the cycle time of CALIPSO is 16 days with removals of cloud cases. Given that the night cloud fraction is âĹij60% over central China (King et al., 2013), the available CALIPSO sampling for matching Lidar is limited. The author should describe the continuous observation period for Lidar, which may be longer than 2-year.

The authors' Answer: Thank you very much for your suggestion and guidance. I am very sorry that we did not describe clearly the time of data. The data collection time was from January 2013 to December 2017, which was only mentioned in the Conclusions. During this time, the total number of CALIPSO crossing Wuhan were 93. After removing the cloud cases, there were 49 valid samples. Moreover, the number of the ground-based Lidar and RS data matching CALIPSO data were 21 and 49, respectively. According to your suggestion, the descriptions about continuous observation period for Lidar and CALIPSO were added in the 2.2 and 2.3 sections.

All the lines and pages indicated above are in the revised manuscript. Thank you for the kind advice.

Sincerely

yours,

Boming Liu

---

## Short Comment (SC2) · 29 Jul 2018

1. This study follows Liu and Liang's method but defines the atmospheric boundary layer in a completely different way. The authors need to verify that the residual layer is the atmospheric boundary layer, and may change the title to avoid misunderstanding.

2. OK, let's assume that RS defines the inversion layer top as the boundary layer height (I am not familiar with "inversion layer"). In this study, the residual layer height at 0200 completely differs from the inversion layer height at 0200, but is highly correlated with the inversion layer height at 2000. I think the authors shouldn't interpret such good result as a "coincidence".

Moreover, the inversion layer top at 2000 considerably differs from the top of mixing

layer in the daytime. In Liu and Liang. (2010), the inversion layer tops at 2000 are ∼500m at all land sites.

This would be my final comment for this paper.

---

## Referee Comment (RC1) · Anonymous Referee #1 · 30 Jul 2018

**General comments:**

3

The algorithm for PBL height retrieval from CALIPSO is still lacking due to the strong influence induced by attenuation of clouds in the PBL and complex meteorology. This manuscript proposed a novel graphic distribution algorithm to derive BLH, which are subject to further validation by comparing with BLHs from collocated radiosonde and ground-based lidar observations. Results are found to be interesting. The findings contribute a lot to the boundary layer community. Overall, this manuscript is well written, and the methodology is also sound. However, there are several issues to be clarified or addressed before it can be accepted for publication in AMT.

**Major points**

1. The biggest concerns of mine is the sounding time for RS is 2000 LT, which is roughly 6 hours before the CALIPSO nighttime overpass at Wuhan. The inter-comparison of BLH between CALIPSO and RS (Fig. 9) seems flawed.  I guess that the authors hypothesize the PBL does not vary considerable over time during nighttime. At the very least, however, the authors should discuss this issue in detail.

2. In section 2 or section 3: Clarification for the averaging scheme for CALIPSO profiles by taking various horizontal smoothing number (i.e., 1, 3, 15 and 30) should be added. Also, to make the results more robust, the horizontal smoothing numbers of 1,3,6,9,12,15, 18 and 30 (i.e., 1/3, 1, 2, 3, 4, 5, 6 and 10km in the along-track direction) are suggested to take. As a result, Fig. 9 can be expanded to take into account more sensitive results.

**Minor points:**

Page 1 Line 17-24:  It will be better to move "The algorithm provided a reliable result when the horizontal smoothing number was greater than 5."  before "This finding indicated…". In addition, what is the logics for the threshold (i.e., 5) of  horizontal smooth number claimed here, since you only analyzed the results by assuming "1, 3, 15 and 30" instead of "5". From my understanding, Figs. 7 and 9 are not enough to draw this conclusion, and thus necessary clarification will be necessary.

Page 1 Line 28-35: The literature review seems in disorder, which can be improved only be rewriting. For example, the authors emphasized twice the role of BLH in environmental health, but I did not find any references supporting it. On top of this issue, the role of PBL is well recognized to be associated with aerosol pollution, which should be mentioned here. Towards this end, the review paper by Li et al, 2017 can be cited here.

Reference:

*Li Z., et al., 2017. Aerosol and boundary-layer interactions and impact on air quality, National Science Review, 4 (6), 810–833. doi: 10.1093/nsr/nwx117.*

Page 2 Line 2: The acronym for "RS" refers to radiosonde? Given its first appearance in this manuscript, its full name should be spelled here.

Page 2 Line 7: …is usually TOO sparse..

Page 2 Line 10: ..can CONTINOUSLY detect..

Page 2 Line 28: Guo et al. 2016 only focuses on the BLH retrieval from radiosonde in China rather than that from satellite measurements. This citation can be replaced with Zhang et al. 2016. Accordingly, Guo et al. 2016a can be considered to move to Page Line 7 "(Seibert et al. 2000; Sawyer et al. 2013; Guo et al., 2016a)".

Page 3 line 9: Liu et al. 2018a is missing in references. The authors can consider citing the following reference here:

Reference:

*Liu, L., et al., 2018a. Elucidating the relationship between aerosol concentration and summertime boundary layer structure in central China. Environmental Pollution 241, 646-653, doi: 10.1016/j.envpol.2018.06.008.*

Page 3 Line 12: not completely coincide WITH ground-based Lidar station ??  How about the distance between CALIPSO track and radiosonde site? The track of CALIPSO shown in Fig.1 should be for the nighttime, which deserves clarification.

Page 3 Line 29: Necessary justification is required for the authors only applying nighttime CALIPSO measurements to estimate BLHs. One reason is that there is higher SNR in nighttime relative to daytime SNR (Winker et al. 2009; Guo et al., 2016b).

Reference:

*Guo, J. et al. , 2016b. Three-dimensional structure of aerosol in China: A perspective from multi-satellite observations, Atmospheric Research, 178–179: 580–589. doi: 10.1016/j.atmosres.2016.05.010.*

---

## Referee Comment (RC2) · Anonymous Referee #2 · 31 Jul 2018

Comments:

The atmospheric boundary layer height is of great importance because it not only affects the diffusion of air pollutions but also determines the formation of different kinds of weather. Previous studies have focused on the determination of BLH based on measurements of remote sensing, especially laser remote sensing. This study proposed an interesting method, which can obtain reliable BLH from CALIPSO data. I have given a review report before, and authors made some revision of the manuscript. Now it looks much better. But I still have some major concerns.

Major comments:

1. In the noise removal phase, how much points were removed in the end? If it is 100

data points, 60 are removed at once and only 40 valid points remain. Can the results be trusted? The authors should add some quality control, such as removing 10 or less, the best quality, 30 are not credible, etc. I did not see the description in the paper.

2. Figure 9, this study shows the comparsion between the BLHs from CALIPSO at 0210LT and RS at 2000LT. But the BLH has strong diurnal variances, this comparison is unreasonable. I suggest that the author change to RS data at night, or delete this comparison.

3. The author claimed that they use nighttime data of CALIPSO and Lidar (0210LT), but the nighttime BLHs at 0210LT from CALIPSO and Lidar looks a little high. It may be due to the that the Lidar system regarded the top of residual layer as the BLH at night. So, the authors should explain it clearly.

4. About data collection time, the authors claimed that the number of residual CALIPSO data over Wuhan area was 49 after removing the cases with cloud and dust. The author should describe the continuous observation period for Lidar and RS, and indicate that how many cases have collected.

5. The principle that satellite data matches the ground station did not appear in the paper. The authors should clarify the match distance range and time range between the CALIPSO and the ground lidar (RS). Because the returns trajectory of CALIPSO is not completely coincident. It is necessary to point out the match distance range and time range.

Technical comments:

1. P2, Line 2: RS is the abbreviation., It should give the full name when it first appears

2. The English of the paper should be improved.

---

## Author Response (AR1)

Dear teacher1:

Thank you very much for your guidance and advice. We carefully read your suggestions, and revised the manuscript in accordance with your comments.

**1. The reviewer's comment: The biggest concerns of mine is the sounding time for RS is 2000LT, which is roughly 6 hours before the CALIPSO nighttime overpass at Wuhan. The inter-comparison of BLH between CALIPSO and RS (Fig. 9) seems flawed. I guess that the authors hypothesize the PBL does not vary considerable over time during nighttime. At the very least, however, the authors should discuss this issue in detail.**

The authors' Answer: Thank you very much for your suggestion and guidance. As your said, due to the time of RS is not matched with the time of CALIPSO, the inter-comparison of BLH between CALIPSO and RS was unreasonable. Another reviewer also pointed out this issue and suggested we delete this comparison. Therefore, we delete the inter-comparison of BLH between CALIPSO and RS to avoid misleading readers. In addition, we increased the inter-comparison of BLH between CALIPSO and Lidar. The horizontal smoothing numbers of 1, 3, 6, 9, 12, and 15 (i.e., 1/3, 1, 2, 3, 4, and 5km in the along-track direction) are add to test the GDM algorithm. It can be seen in P, line (Fig.9).

**2. The reviewer's comment: In section 2 or section 3: Clarification for the averaging scheme for CALIPSO profiles by taking various horizontal smoothing number (i.e., 1, 3, 15 and 30) should be added. Also, to make the results more robust, the horizontal smoothing numbers of 1, 3, 6, 9, 12, 15, 18 and 30 (i.e., 1/3, 1, 2, 3, 4, 5, 6 and 10km in the along-track direction) are suggested to take. As a result, Fig. 9 can be expanded to take into account more sensitive results.**

The authors' Answer: Thank you very much for your suggestion and guidance. Due to the section 3 was used to describe the process of the GDM algorithm, we did not add

the various horizontal smoothing number (i.e., 1, 3, 15 and 30). According to your suggestion, the horizontal smoothing numbers of 1, 3, 6, 9, 12, and 15 (i.e., 1/3, 1, 2, 3, 4, and 5km in the along-track direction) are add to test the GDM algorithm. The new Fig.9 was shown below. Due to the correlation coefficient tends to be stable when the horizontal smoothing numbers was 12 and 15. So we did not analyze the comparison results when the horizontal smoothing numbers was 18 and 30.

[Figure]

The modification can be seen in the P6, line37-40 and P7, line 1-6. "Fig. 9 show the correlation coefficients between the BLH derived from CALIPSO and ground-based Lidar under the horizontal smoothing numbers of 1, 3, 6, 9, 12 and 15. The red and blue points represent the BLH calculated by GDM algorithm and MSD method, respectively. Figs. 9a, 9b and 9c show the comparison of BLH between CALIPSO and Lidar under the horizontal smoothing number of 1, 3 and 6. The correlation coefficients between the BLH derived by GDM algorithm and ground-based Lidar were 0.12, 0.14 and 0.47, respectively. Meanwhile, the correlation coefficients between the BLH derived by MSD method and ground-based Lidar were 0.1, 0.27 and 0.33. Figs. 9d, 9e and 9f show the comparison of BLH between CALIPSO and Lidar under the horizontal smoothing number of 9, 12 and 15. The correlation coefficients between the BLH derived by GDM algorithm and Lidar measurements were both 0.72, and the correlation coefficients between the BLH derived by MSD method and Lidar measurements were 0.54, 0.62 and 0.7, respectively."

**3. The reviewer's comment: Page 1 Line 17-24: It will be better to move "The algorithm provided a reliable result when the horizontal smoothing number was greater than 5." Before "This finding indicated…". In addition, what is the logics for the threshold (i.e., 5) of horizontal smooth number claimed here, since you only analyzed the results by assuming "1, 3, 15 and 30" instead of "5". From my understanding, Figs. 7 and 9 are not enough to draw this conclusion, and thus necessary clarification will be necessary.**

The authors' Answer: Thank you very much for your suggestion and guidance. According to your suggestion, we move the sentence to the specified location. In addition, we did more experiments and reanalyzed Fig.9. Based on the new results, the GDM algorithm can provide a reliable result when the horizontal smoothing number was greater than 9. Therefore, we modified the descriptions in the P1, line 23-25. "The algorithm provided a reliable result when the horizontal smoothing number was greater than 9. This finding indicated that the proposed algorithm can be applied to the CALIPSO satellite data with 3 and 5 km horizontal resolution."

**4. The reviewer's comment: Page 1 Line 28-35: The literature review seems in disorder, which can be improved only be rewriting. For example, the authors emphasized twice the role of BLH in environmental health, but I did not find any references supporting it. On top of this issue, the role of PBL is well recognized to be associated with aerosol pollution, which should be mentioned here. Towards this end, the review paper by Li et al, 2017 can be cited here.**

The authors' Answer: Thank you very much for your suggestion and guidance. According to your suggestion, we rewrite the literature review in the P1, line 30-35. Moreover, the review paper by Li et al, 2017 was add in P1, Line 35. "Therefore, the boundary layer height (BLH) is essential to atmospheric aerosol pollution and must be accurately and continuously monitored (Li et al. 2017)." "Li, Z., Guo, J., Ding, A., Liao, H., Liu, J., Sun, Y., ... & Zhu, B. (2017). Aerosol and boundary-layer interactions and impact on air quality. National Science Review, 4(6), 810-833."

**5. The reviewer's comment: Page 2 Line 2: The acronym for "RS" refers to radiosonde? Given its first appearance in this manuscript, its full name should be spelled here.**

The authors' Answer: Thank you very much for your suggestion and guidance. In here, the RS refers to radiosonde. According to your suggestion, its full name was given in the P2, line 3.

**6. The reviewer's comment: Page 2 Line 7: …is usually TOO sparse...**

The authors' Answer: Thank you very much for your suggestion and guidance. According to your suggestion, we add the "too" in the P2, line 8. "Moreover, the spatial coverage of RS sites is usually too sparse to capture BLH spatial variability."

**7. The reviewer's comment: Page 2 Line 10: ...can CONTINOUSLY detect...**

The authors' Answer: Thank you very much for your suggestion and guidance. According to your suggestion, we add the "continuously" in the P2, line 11. "Lidar systems can continuously detect the BLH from the aerosol vertical profile."

**8. The reviewer's comment: Page 2 Line 28: Guo et al. 2016 only focuses on the BLH retrieval from radiosonde in China rather than that from satellite measurements. This citation can be replaced with Zhang et al. 2016. Accordingly, Guo et al. 2016a can be considered to move to Page Line 7 "(Seibert et al. 2000; Sawyer et al. 2013; Guo et al., 2016a)"**

The authors' Answer: Thank you very much for your suggestion and guidance. According to your suggestion, this citation was replaced with Zhang et al. 2016. Moreover, Guo et al. 2016a was moved to P2, Line 1. "(Seibert et al. 2000; Sawyer et al. 2013; Guo et al. 2016a)"

**9. The reviewer's comment: Page 3 line 9: Liu et al. 2018a is missing in references. The authors can consider citing the following reference here:**

The authors' Answer: Thank you very much for your suggestion and guidance. According to your suggestion, we add the reference in the P3, line 4. "Liu, L., Guo, J., Miao, Y., Li, J., Chen, D., He, J., & Cui, C. (2018c). Elucidating the relationship between aerosol concentration and summertime boundary layer structure in central China. Environmental Pollution, 241, 646-653."

**10. The reviewer's comment: Page 3 Line 12: not completely coincide WITH ground-based Lidar station? How about the distance between CALIPSO track and radiosonde site? The track of CALIPSO shown in Fig.1 should be for the nighttime, which deserves clarification.**

The authors' Answer: Thank you very much for your suggestion and guidance. About the matching principles of ground-based Lidar and CALIPSO, we have explained it in two aspects. First, the distance between CALIPSO and ground-based Lidar stations is within 50 km. Moreover, the ground-based Lidar data were obtained within 30 min of CALIPSO overpass times. According to your suggestion, we add the descriptions in in the P3, line "7". "About matching principles of ground-based and space-borne Lidar, the distance between CALIPSO and ground-based Lidar stations is within 50 km. Meanwhile, the ground-based Lidar data were obtained within 30 min of CALIPSO overpass times."

**11. The reviewer's comment: Page 3 Line 29: Necessary justification is required for the authors only applying nighttime CALIPSO measurements to estimate BLHs. One reason is that there is higher SNR in nighttime relative to daytime SNR (Winker et al. 2009; Guo et al., 2016b).**

The authors' Answer: Thank you very much for your suggestion and guidance. As your said, there is higher SNR in nighttime relative to daytime SNR. According to your suggestion, we add some descriptions in the P3, line 28-29. "Due to the nighttime data have a higher SNR relative to daytime data (Winker et al. 2009; Guo et al., 2016b)."

Many grammatical or typographical errors have been revised.

All the lines and pages indicated above are in the revised manuscript. Thank you for the kind advice.

Sincerely

yours, Boming Liu

Dear teacher2:

Thank you very much for your guidance and advice. We carefully read your suggestions, and revised the manuscript in accordance with your comments.

**1. The reviewer's comment: In the noise removal phase, how much points were removed in the end? If it is 100 data points, 60 are removed at once and only 40 valid points remain. Can the results be trusted? The authors should add some quality control, such as removing 10 or less, the best quality, 30 are not credible, etc. I did not see the description in the paper.**

The authors' Answer: Thank you very much for your suggestion and guidance. In the last review comment reply, we have explained this point. In fact, the noise point is not eliminated under the noise removal phase, but is judged as a cluster which same with the neighboring particles. In this way, it won't lose height information. Meanwhile, the class misjudgment caused by noise point is corrected. Therefore, in the noise removal phase, it does not need to add quality control. But this point did not explain clearly in the manuscript. To avoid misleading readers, we add some descriptions in the P5 line 3. "According to the noise removal principle, the category of noise point was judged as a cluster which same with the neighboring particles."

**2. The reviewer's comment: Figure 9, this study shows the comparison between the BLHs from CALIPSO at 0210LT and RS at 2000LT. But the BLH has strong diurnal variances, this comparison is unreasonable. I suggest that the author change to RS data at night, or delete this comparison.**

The authors' Answer: Thank you very much for your suggestion and guidance. As your said, due to the time mismatch, the comparison between the BLHs from CALIPSO at 0210LT and RS at 2000LT was unreasonable. So according to your suggestion, we delete the comparison between CALIPSO and RS. In addition, to make the results more robust, more CALIPSO profiles by different horizontal smoothing number was added

in Fig.9. The horizontal smoothing numbers of 1, 3, 6, 9, 12, and 15 (i.e., 1/3, 1, 2, 3, 4, and 5km in the along-track direction) are add to test the GDM algorithm, as the following picture shown.

[Figure]

**3. The reviewer's comment: The author claimed that they use nighttime data of CALIPSO and Lidar (0210LT), but the nighttime BLHs at 0210LT from CALIPSO and Lidar looks a little high. It may be due to the that the Lidar system regarded the top of residual layer as the BLH at night. So, the authors should explain it clearly.**

  The authors' Answer: Thank you very much for your suggestion and guidance. As your said, the structure of boundary layer is divided into a stable layer and a residual layer in the nighttime. The Lidar system obtained the boundary layer height based on the aerosol scattering profile. If the aerosol loading in the residual layer is large, the top of residual layer would be identified as the boundary layer height by Lidar. After our experiment, we found that the CALIPSO system was hard to identify the top of stable layer in nighttime. Therefore, the top of residual layer was defined as the boundary layer height in CALIPSO and Lidar system. It leads to that the BLHs from CALIPSO and Lidar are all a little high. About this question, more details would be added in the 3.2 section (Error analysis) to avoid misleading readers. Meanwhile, overcoming the effect of the residual layer on CALIPSO is our future work. According to your

suggestion, we add some descriptions in the P5, line 23-24. "In addition, due to the effect of the nocturnal residual layer, the top of residual layer would be identified as the BLH by Lidar system in some cases."

**4. The reviewer's comment: About data collection time, the authors claimed that the number of residual CALIPSO data over Wuhan area was 49 after removing the cases with cloud and dust. The author should describe the continuous observation period for Lidar and RS, and indicate that how many cases have collected.**

The authors' Answer: Thank you very much for your suggestion and guidance. I am very sorry that we did not describe clearly the time of data. The experimental time was from January 2013 to December 2017. During this time, the total number of CALIPSO crossing Wuhan were 93. After removing the cloud cases, there were 49 valid samples. Moreover, the ground-based Lidar and RS data were collected at the same time. The number of the ground-based Lidar and RS data matching CALIPSO data were 21 and 49, respectively. According to your suggestion, the descriptions about continuous observation period for Lidar and CALIPSO were added in the P3, line 19-21 and 30-33. "The Lidar data was collected from January 2013 to December 2017. After matching the CALIPSO data, the valid number of the ground-based Lidar data were 21 cases." "The data collection time was from January 2013 to December 2017. During this time, the total number of CALIPSO crossing Wuhan were 93. After removing the cloud cases, there were 49 valid samples."

**5. The reviewer's comment: The principle that satellite data matches the ground station did not appear in the paper. The authors should clarify the match distance range and time range between the CALIPSO and the ground lidar (RS). Because the returns trajectory of CALIPSO is not completely coincident. It is necessary to point out the match distance range and time range.**

The authors' Answer: Thank you very much for your suggestion and guidance. About the matching principles of ground-based Lidar and CALIPSO, we have explained it in two aspects. First, the distance between CALIPSO and ground-based Lidar stations is within 50 km. Moreover, the ground-based Lidar data were obtained within 30 min of CALIPSO overpass times. According to your suggestion, we add the descriptions in in the P3, line 7-10. "About matching principles of ground-based and space-borne Lidar, the distance between CALIPSO and ground-based Lidar stations is within 50 km. Meanwhile, the ground-based Lidar data were obtained within 30 min of CALIPSO overpass times."

**6. The reviewer's comment: P2, Line 2: RS is the abbreviation. It should give the full name when it first appears.**

The authors' Answer: Thank you very much for your suggestion and guidance. In here, the RS refers to radiosonde. According to your suggestion, its full name was given in the P2, line 3.

**7. The reviewer's comment: The English of the paper should be improved.**

The authors' Answer: The authors' Answer: Thank you very much for your patience and guidance. I am very sorry for my poor English expression. To improve the poor language, I have get a professional language editing service to correct the language.

Many grammatical or typographical errors have been revised.

All the lines and pages indicated above are in the revised manuscript. Thank you for the kind advice.

Sincerely

yours, Boming Liu

[revised manuscript text omitted]

**2.3 CALIPSO Data**

The CALIOP satellite is the first space-borne Lidar optimised for aerosol and cloud profiling, which has 532 nm channel (parallel and perpendicular polarisation) and 1064 nm channel (Liu et al. 2009). This satellite can provide the total attenuated backscatter coefficient 532 and attenuated backscatter coefficient 1064 with a horizontal resolution of 1/3 km and vertical resolution of 30 m. Attenuated backscatter data (Level 1B) were used for testing the proposed algorithm. The cycle time of CALIPSO across the central China region is 16 days, and the crossing time of the satellite in Wuhan is 13:10 and 02:10 local time. Due to the nighttime data have a higher SNR relative to daytime data (Winker et al. 2009; Guo et al., 2016b). The nighttime data were employed for this analysis for the matching of the ground-based data, and cases with cloud and dust were removed in this study. The data collection time was from January 2013 to December 2017. During this time, the total number of CALIPSO crossing Wuhan were 93. After removing the cloud cases, there were 49 valid samples.

*2.4 Radiosonde Measurements Data*

~~The RS data were provided by the Bureau of Meteorology at Wuhan site, which is 23 m above sea level and 30 km northwest from the Lidar site. The RS was launched twice a day at 8:00 (LT) and 20:00 (LT). The RS data from 20:00 (LT) were selected to calculate the BLH and match the satellite data (Pal et al. 2013). The vertical profiles of the mean horizontal wind speed and potential temperature were used to determine the BLH following the method described in Liu and Liang because the construction of nighttime boundary layer is complicated (Liu et al. 2010). Moreover, due to the mismatched time of RS data, the BLH estimated from RS measurements data cannot be regarded as 'truth'; thus, the estimated BLH is jointly used with the ground-based Lidar for validating CALIPSO results.~~

**3. Methodology**

**3.1. Method**

Previous studies reported that the different particles are distributed in different vertical heights (Liu et al. 2018c; Sugimto et al. 2002). Most of the particles above the boundary layer are molecular particles, and the particles below the boundary layer are mainly aerosol particles, as shown in Figs. 2a and 2c, respectively. Therefore, we proposed a dual-wavelength algorithm that determines BLH on the basis of two-dimensional graphical distribution. The total attenuated backscatter coefficient 532 ($TAB_{532}$) and attenuated backscatter coefficient 1064 ($AB_{1064}$) were used to construct the two-dimensional graphical distribution. The specific steps are as follows:

Firstly, the $TAB_{532}$ and $AB_{1064}$ were employed for the construction of the sample sequence $X(z)$. As shown in Figs. 2b and 2d, the $TAB_{532}$ and $AB_{1064}$ represent the aerosol vertical profile at 532 and 1064 wavelength measured by CALIPSO, respectively. The $X(z)$ can be expressed as:

$$\left[ X(z) \right] = \left[ TAB_{532}(z), AB_{1064}(z) \right]$$ (1)

where $z$ stands for the altitude of sample points; $X(z)$ represents the coordinates of the sample point at the altitude of z; $TAB_{532}(z)$ and $AB_{1064}(z)$ represent the total attenuated backscatter (532 nm) and attenuated backscatter (1064 nm) value of the sample point at the altitude of z, respectively.

The sample sequence $X(z)$ is shown in Fig. 3a. The colour bar is the altitude of sample points. The figure shows that $TAB_{532}$ and $AB_{1064}$ of blue points (the particles below the boundary layer) were larger than those of the red points (the particles above the boundary layer). According to this two-dimensional distribution, the sample sequence $X(z)$ can be divided into two categories.

The k-means method was used for the classification of the sample sequence. Two centroid points (u1, u2) were randomly selected from the sample sequence. For each sample point of sample sequence $X(z)$, the cluster C belonging to is calculated as follows:

$$c(z) = \arg \min_{j} \left\| X(z) - u_j \right\|^2$$ (2)

where $C(z)$ represents the cluster of sample point at the altitude of z, and $u_j$ is the centroid of cluster j ($u_1$ or $u_2$). For each cluster j, the centroid $u_j$ is recalculated as follows:

$$u_j = \frac{\sum_{i=1}^{m} 1\{c(i) = j\} x(i)}{\sum_{i=1}^{m} 1\{c(i) = j\}} \tag{3}$$

Eqs. (3) and (4) are repeated until the centroids ($u_1$ and $u_2$) converge. The sample sequence is divided into two categories after the convergence. As shown in Fig. 3b, cluster$_2$ (blue points) indicates the aerosol particles below the boundary layer, and cluster$_1$ (red points) is the molecular particles above the boundary layer. Black cross represents centroid points. Meanwhile, the categories sequence $f(z)$, which changes with height, can be obtained and expressed as:

$$f(z) = \begin{cases} 1, & z \in cluster_1 \\ 2, & z \in cluster_2 \end{cases} \tag{4}$$

where $f(z)$ is the category of sample point at the altitude of $z$. The noise points would affect the classification results due to the large noise of satellite data. Therefore, the noise points on the categories sequence must be eliminated. The noise point was determined by comparing two points near the point. If the two points above and below this point belong to the same class, then this point should also belong to this category. The noise point can be filtered by:

$$f(z) = f(z-m), \quad if : f(z-m) = f(z+m) \tag{5}$$

where $m$ represents the multiple of the vertical resolution, and the different values can be selected at different noise levels. When the horizontal smoothing number is small and the signal noise is large, the value of $m$ can be set as 2; and when the horizontal smoothing number is large, the value of $m$ can be set as 1. According to the noise removal principle, the category of noise point was judged as a cluster which same with the neighbouring particles. Hence, the noise points were removed, and the new categories sequence $F(z)$ was obtained as follows:

$$F(z) = \begin{cases} 1, & z > BLH \\ 2, & z < BLH \end{cases} \tag{6}$$

where $F(z)$ represents the category of the sample point at the altitude of $z$. $BLH$ indicates the BLH result. Fig. 3c shows the category sequence $F(z)$, which contains the height information and shows evident variation at the top of boundary layer. Therefore, the maximum gradient of the categories sequence $F(z)$ is the top point of boundary layer. The $BLH$ can be calculated by searching the maximum gradient, which can be expressed as:

$$BLH = \left| d\left[ F(z) \right] \right|_{max} \tag{7}$$

Following this process, the BLH was obtained based on the two-dimensional distribution of particles.

**3.2. Error analysis**

Fig. 4 shows the flowchart of the GDM algorithm. Four calculation steps are available: establishing the sample sequence, particle clustering, filtering noise points, and maximum gradient searching. The error of input parameters is the main factor affecting the accuracy of the algorithm because these steps are quantitative calculations. According to the official description, the uncertainty of backscatter coefficient was 20%−30% (Winker et al. 2009). The total attenuated backscatter at 532 nm wavelength and the attenuated backscatter at 1064 nm wavelength were measured from CALIPSO. Therefore, the error of input parameters was 20%−30%. The error

of the BLHs derived by the GDM algorithm is approximately 20%–30%. In addition, it need to note that this method cannot be applied to low cloud and dust cases, because the boundary of cloud or dust would be misclassified to BLH. In addition, due to the effect of the nocturnal residual layer, the top of residual layer would be identified as the BLH by Lidar system in some cases.

[revised manuscript text omitted]

Guo, J., Liu, H., Wang, F., Huang, J., Xia, F., Lou, M., ... & Yung, Y. L. (2016b). Three-dimensional structure of aerosol in China: A perspective from multi-satellite observations. Atmospheric Research, 178, 580-589.

Hennemuth, B., & Lammert, A. (2006). Determination of the atmospheric boundary layer height from radiosonde and lidar backscatter. Boundary-Layer Meteorology, 120(1), 181-200.

Holzworth, G. (1964), Estimates of mean maximum mixing depths in the contiguous United States, Mon. Weather Rev., 92(5), 235–242.

Holzworth, G. (1967), Mixing depths, wind speeds and air pollution potential for selected locations in the United States, J. Appl. Meteorol., 6(6), 1039–1044.

Huang, Z., Huang, J., Bi, J., Wang, G., Wang, W., Fu, Q., ... & Shi, J. (2010). Dust aerosol vertical structure measurements using three MPL lidars during 2008 China-US joint dust field experiment. Journal of Geophysical Research: Atmospheres, 115(D7).

Jordan, N. S., Hoff, R. M., & Bacmeister, J. T. (2010). Validation of Goddard Earth Observing System-version 5 MERRA planetary boundary layer heights using CALIPSO. Journal of Geophysical Research: Atmospheres, 115(D24).

Liu, B., Ma, Y., Gong, W., Zhang, M., & Yang, J. (2018a). Determination of boundary layer top on the basis of the characteristics of atmospheric particles. Atmospheric Environment, 178, 140-147.

Leventidou, E., P. Zanis, D. Balis, E. Giannakaki, I. Pytharoulis, and V. Amiridis (2013), Factors affecting the comparisons of planetary boundary layer height retrievals from CALIPSO, ECMWF and radiosondes over Thessaloniki, Greece, Atmos. Environ., 74, 360–366.

Lange, D., Tiana-Alsina, J., Saeed, U., Tomas, S., & Rocadenbosch, F. (2014). Atmospheric boundary layer height monitoring using a Kalman filter and backscatter lidar returns. IEEE transactions on geoscience and remote sensing, 52(8), 4717-4728.

Li, H., Yang, Y., Hu, X. M., Huang, Z., Wang, G., Zhang, B., & Zhang, T. (2017). Evaluation of retrieval methods of daytime convective boundary layer height based on lidar data. Journal of Geophysical Research: Atmospheres, 122(8), 4578-4593.

Liu, B., Ma, Y., Gong, W., Zhang, M., & Yang, J. (2018b). Study of continuous air pollution in winter over Wuhan based on ground-based and satellite observations. Atmospheric Pollution Research, 9(1), 156-165.

Liu, L., Guo, J., Miao, Y., Li, J., Chen, D., He, J., & Cui, C. (2018c). Elucidating the relationship between aerosol concentration and summertime boundary layer structure in central China. Environmental Pollution, 241, 646-653.

[revised manuscript text omitted]